# Effects of Carbon Black on Mechanical Properties and Oil Resistance of Liquid Silicone Rubber

**DOI:** 10.3390/polym16070933

**Published:** 2024-03-28

**Authors:** Beom-Joo Lee, Hyeong-Min Yoo

**Affiliations:** School of Mechanical Engineering, Korea University of Technology and Education (KOREATECH), Cheonan 31253, Republic of Korea; gadc336@koreatech.ac.kr

**Keywords:** liquid silicone rubber, carbon black, mechanical properties, swelling, oil resistance

## Abstract

Liquid silicone rubber (LSR) garners attention across a diverse range of industries owing to its commendable fluidity and heat resistance. Nonetheless, its mechanical strength and oil resistance fall short compared to other rubbers, necessitating enhancement through the incorporation of a suitable filler. This research focuses on reinforcing LSR using carbon black (CB) particles as a filler, evaluating the mechanical properties and oil resistance of neat LSR, and LSR containing up to 3 wt% of CB filler. CB was added in powder form to investigate its effect on LSR. When LSR was impregnated with oil, the deterioration of rubber was noticeably observed under high-temperature conditions compared to room-temperature conditions. Consequently, the mechanical properties and oil resistance, excluding the permanent compression reduction rate, tended to increase as the filling content of CB increased compared to the unfilled state. Notably, in the specimen with 2 wt% CB filler, the tensile modulus increased significantly by 48% and the deterioration rate was reduced by about 50% under accelerated deterioration conditions. Additionally, the swelling rate in oil decreased by around 14%. This validates a notable improvement in both mechanical properties and oil resistance. Based on the identified mechanism for properties enhancement in this study, CB/LSR composite is expected to have a wide range of applications in fields such as gaskets, oil seals, and flexible sensors.

## 1. Introduction

Liquid silicone rubber (LSR) has excellent biocompatibility and thermal compatibility, and its good fluidity makes it easy to implement into products, so it can be used even in places where complex shapes and strict deviations are required [1,2,3,4,5]. Due to these characteristics, LSR is widely used and attracts attention in various applications, such as medicine [6], electronics [7,8], and automobiles [9]. However, LSR’s mechanical strength and oil resistance are relatively inferior to those of other rubbers, which are directly related to product performance and limit its widespread application. In particular, when LSR is used in products usually exposed to oil, such as gaskets, oil seals, and O-rings, compression set behavior that maintains elastic properties even under long-term compressive stress, temperature, and swelling ratio for oil must be considered. Corrosion or deterioration which occur due to the oil exposure reduce the performance and lifespan of the products, so research on processing the silicone rubber with improved mechanical properties and oil resistance is required [10,11,12,13,14].

LSR is mainly composed of long divinyl polydimethylsiloxane chains and short SiH-multifunctionalized copolymers, along with silica, platinum catalysts, and cure retarders. It is usually supplied in two parts, Part A and Part B, one containing the catalyst and the other without, and is used in a 1:1 ratio. The curing process of LSR involves hydrosilylation, wherein Si-H bonds are added to the carbon double bonds of the polydimethylsiloxane chain. For stability and compatibility with silicon, a platinum catalyst is employed, and a cure retarder is used to prevent premature crosslinking of the polymer at room temperature. The platinum-mediated crosslinking reaction then takes place at elevated temperatures. Since LSR products are processed from A and B solutions, it has the advantage that reinforcing materials can be added during the process. Therefore, Consequently, fillers like silica, clay, graphene oxide, carbon nanotubes (CNTs), and carbon black can be readily added to enhance the properties of LSR [2,3,4]. 

Carbon-based reinforcement materials such as carbon black (CB), CNT, and graphene have high thermal conductivity and chemical stability, so they are most commonly used to produce various elastomer composites for the purpose of reinforcing conductivity and mechanical properties. Among them, CB in particular has the advantage of a low price, and it is known to have a large specific surface area and a chemically active surface in a complex microstructure, which has a rich reaction with rubber molecular chains [15,16,17,18]. ZH Li et al. [15] added five types of carbon black with different surface areas to EPDM rubber and confirmed that physical and chemical crosslinking occurred most actively in CB with the largest surface area compared to other CBs. In addition, it had the best hardness, tensile strength, and 300% elastic modulus, but showed relatively low elongation. N. Stübler et al. [16] confirmed the mechanical stress of CB-containing Elastomer (EPDM, SBR, NR) through uniaxial deformation cycle tests and, simultaneously, DC (direct current) resistance of composite samples was investigated under deformation conditions. The effects of strain and relaxation on the carbon black network of the elastomer were confirmed through the investigated stress and piezo resistance. A. Mostafa et al. [10] investigated the mechanical properties and oil resistance performance of CB-reinforced SBR and NBR rubber according to time, temperature, and CB content. It was revealed that as the CB content increases, the swelling rate with exposure to the oil decreases, and as the exposure time and temperature to the oil increase, the oil penetrates between the rubber chains and the swelling rate increases. In most existing studies, CB is added in micro and nano forms to ensure uniform dispersion. When adding CB, if the amount is excessive, agglomeration between fillers may occur and the mechanical properties of the material would not be uniform, so it is necessary to consider the amount of filler for even dispersion [19,20,21,22,23,24].

Also, the addition of carbon-based reinforcements enhances the electrical properties of flexible materials, making them applicable to the electrical industry. As the electronics and electrical equipment industry evolves, LSR composites with CB are utilized in a variety of applications for sensing or electrical detection purposes, act as shielding for electromagnetic interface (EMI), and improve the durability of medical and automotive components where LSRs have traditionally been used. Pin Liu et al. [25] designed a pressure-/temperature-sensitive conductive silicone rubber filled with different volume fractions of CB, prepared by a liquid mixing method. Pan Song et al. [26] designed flexible sensors made of silicone rubber/graphene/modified CB composites, which are highly likely to be applied in the wearable electronic devices field. Javad Jeddi et al. [27] investigated the electrical conductivity and EMI shielding properties of silicon rubber, CB, nanographite, and polyurethane foam hybrid composites.

Previous studies on CB/LSR composites have focused on specific properties depending on the application, such as thermal conductivity, thermal stability, conductivity, and electrical resistance [28,29,30,31]. Zhang et al. [28] measured the activation energy through thermal decomposition by adding different weights of conductive CB to silicone rubber and confirmed that conductive silicone rubber had superior thermal stability compared to general silicone rubber, and Ding et al. [29] produced a CB-filled silicone rubber composite, and confirmed that both the stress and electrical resistance of the specimen increased immediately after compression, and that the stress relaxed and gradually decreased, resulting in the use of a flexible sensor that can measure compressive stress relief.

Although research is actively being conducted on other characteristics of LSR containing CB, there is a lack of quantitative research on oil resistance. Considering the diverse applications of LSR, there is still a need to investigate the oil resistance and extensive mechanical properties of CB-reinforced LSR. Accordingly, in this study, we fabricated LSR reinforced with CB of various concentrations, and performed a compression test, tensile test, swelling test, and compression set test to determine the effects of CB on the oil resistance of LSR.

## 2. Experimental Section

### 2.1. Preparation of CB/LSR Composite Specimen

KCC Corp’s SL7270 (Seoul, Republic of Korea) product was used as LSR, and Orion Engineered Carbon’s No. 0924795 (Senningerberg, Luxembourg) product was used for CB as a rubber reinforcement. To produce specimens, CB was dispersed in the LSR part B solution, which included vinyl polymer, silica, H polymer, and a curing retardant. The part B solution was then mixed with the LSR part A solution for 10 min by using a mechanical stirrer [23,24], consisting of vinyl polymer, silica, and platinum catalyst, in a mixing ratio of 1:1. Subsequently, using the Automatic Heating Plate, QM900A from Qmesys Corp (Uiwang-si, Republic of Korea), a specimen was manufactured by pressurizing it at 20 bar for 10 min at a temperature of 170 °C (Figure 1). The curing parameters were optimized according to the KCC LSR 7270 data sheet.

### 2.2. Measurements

#### 2.2.1. Tensile and Compression Tests

The tensile test adhered to the ASTM D412 standard [32], and a 1-ton UTM (TestOne, T0-102, Siheung-si, Republic of Korea) was employed for the testing. The crosshead’s test speed was set at 500 mm/min. The specimen size was 40 × 25 mm, and appropriate gripping was applied on both sides of the gripper in the equipment to prevent sample slipping. Tensile tests were conducted in two cases. First, tests were conducted according to the CB content at room temperature, and second, specimens were deteriorated for 72 h in an oil at 130 °C and then measured to determine their CB content.

The compression test also utilized the same UTM as the tensile test, with testing procedures referencing the ASTM D1229 standard [33]. The crosshead test speed was set at 20 mm/min and conducted at room temperature.

#### 2.2.2. Swelling Test

Deterioration needs to be taken into account in the lifespan of rubber products such as gaskets and seals when exposed to an environment containing oil. The deterioration can be quantified as the swelling ratio of rubber exposed to oil under specific temperature and time conditions. In this study, motor oil was chosen and used to simulate the environment surrounding gaskets and seals. The swelling test followed the ASTM D471 standard [34] and was conducted using motor oil at room-temperature and high-temperature conditions (100, 130 °C). The specimen size was 25 × 50 × 2 mm (Figure 2). The specimen underwent immersion for 3 days at room temperature and in an oven (Labtech, LDO-150F, Namyangju-si, Republic of Korea) at 100 °C and 130 °C, and the change in swelling ratio over time was examined. When measuring, the specimen was washed in acetone, wiped with filter paper, and then weighed. In the case of the high-temperature swelling tests, the specimen at high temperature was transferred to the room-temperature motor oil for cooling, followed by washing in acetone and wiping with filter paper. The formula for calculating the swelling ratio is as follows.
(1)$$Q\%=\frac{M_{t}-M_{0}}{M_{0}}\times 100\;$$
$$\left(M_{t}=Mass\;after\;swelling,\;\;M_{0}=initial\;mass\right)$$

#### 2.2.3. Scanning Electron Microscopy (SEM) Measurement

The SEM measurement was conducted using COXEM’s EM-30N series SEM, and multiple images were presented on a computer through field emission scanning electrons. In this experiment, the LSR samples, considering LSR’s status as a non-conductive material incapable of reflecting electrons, were coated with platinum. High-quality SEM images were achieved by directing electrons onto a thin platinum film applied to the sample, and images were captured at 7 kV with a working distance ranging from 4.0 mm to 5.0 mm.

#### 2.2.4. Compression Set Test

The ability of rubber compounds to retain elasticity under prolonged compressive stress is crucial for products demanding long-term performance, such as gaskets and seals. This characteristic can be assessed through a compression set test. The formula for calculating the permanent compression set rate (C%) is as follows.
(2)$$C\%=\frac{T_{0}-T_{1}}{T_{0}-T_{s}}\times 100$$

($T_{0}=$
*Initial thickness of Specimen*,
$T_{1}=$
*Thickness of specimen after clamp removal*,
$T_{s}=$
*Thickness between clamps*).

As evidenced by the above equation, the lower the compression set, the more resistant the material is to permanent deformation at a given deflection and temperature. The compression set test adhered to the ASTM D395 standard [35], employing a cylindrical compression-molded specimen with a diameter of 25 mm and a thickness of 12 mm. As shown in Figure 3, a specimen was inserted between the tools, and the bolt was tightened to narrow the space of the plates. The space between the plates was adjusted to achieve a compression percentage of 25% of the original thickness of the specimen. The final thickness was measured 30 min after loosening the bolt and removing the clamp. The tests were conducted at room temperature and 130 °C.

## 3. Results and Discussion

### 3.1. Tensile and Compression Properties

Through tensile and compressive strength tests, the impact of temperature and the addition of CB on the mechanical properties of LSR was determined. According to the compression test results in Figure 4a, the compressive modulus of the initial elastic deformation section was 40.86 MPa in neat LSR, 44.23 MPa in 1 wt% of CB, 50.72 MPa in 2 wt% of CB, and 54.21 MPa in 3 wt% of CB. The compression modulus of LSR tended to increase with the increasing CB content. From the graphs, it is evident that the compressive behavior of each composite is not consistently uniform throughout. It is apparent that they initially exhibit linear elasticity, but as the deformation progresses beyond a critical value, there is an irregular nonlinear deformation regime, likely attributable to local network destruction. Nevertheless, it has been noted that this phenomenon does not significantly alter the initial slope of the graph, namely, the modulus [31,36,37]. In Figure 4b, the result of the tensile test revealed that as the CB content increased, the elongation of LSR decreased, but the tensile modulus tended to increase. Both the compression and tensile test results indicate that CB contributes to the enhancement of modulus, consistent with previous studies [16,17,18]. In Figure 4c, the specimens were exposed to the same accelerated deterioration condition (130 °C, 72 h) to observe deterioration according to CB content. The specimens with CB had relatively better resistance to deterioration compared to the neat specimen, which shows that the deterioration of mechanical properties may progress more slowly under extreme conditions due to the added CB. The deterioration rates of mechanical properties, specifically ultimate tensile strength and tensile modulus, for each specimen are summarized and presented in Table 1 and Table 2. Notably, for the specimen with 2 wt% CB added, the ultimate tensile strength was shown to be 50% less and the modulus to be 67% less deteriorated compared to the neat LSR specimen. Figure 4d shows that when the specimens were immersed in oil, the decrease in tensile strength and elongation of the specimen accelerated and proceeded under high-temperature conditions compared to low-temperature conditions.

### 3.2. Swelling Characteristics

Swelling tests were conducted at various oil exposure times and temperatures to determine the effect of CB content on the swelling behavior of LSR. Table 3 shows the change in weight corresponding to the oil exposure time of LSR containing 0 to 3 wt% CB at various temperatures of RT, 100 °C, and 130 °C, respectively. As time passed, the oil permeated the specimen, increasing its weight. The change in swelling rate was also calculated using Equation (1) in Section 2.2.2, as shown in Figure 5 and Table 4. Each specimen exhibited a generally similar pattern, wherein the oil absorption initially increases rapidly but subsequently decreases over time, leading to a reduction in the slope of the graphs. The swelling tests revealed a swelling ratio of 5.21% at room temperature, 9.02% at 100 °C, and 11.11% at 130 °C for neat LSR. Compared to the neat LSR results, the swelling ratio decreased as the CB content in LSR increased up to 2 wt%, and this effect was particularly pronounced at high temperatures. The findings show that the CB has a mitigating effect on oil-induced deterioration in LSR. However, as the CB content increased, the swelling ratio decreased, and it was observed that the swelling ratio increased slightly when 3 wt% of CB was added, which is presumed to be due to the agglomeration of the CB [38,39].

Furthermore, it was verified that the variation in swelling rate, dependent on the CB content, becomes more pronounced at elevated temperatures compared to room temperature. This observation indicates that with increasing temperature, the mobility of the rubber chains increases within the viscoelastic behavior of the rubber. Consequently, the available space for oil diffusion between rubber molecules increases, facilitating a more pronounced rubber expansion.

### 3.3. Morphological and Surface Characterization

SEM is a commonly employed method for observing the morphology of rubber/CB composites. The SEM measurement results are shown in Figure 6. We captured SEM images of the surfaces of both Neat LSR and the CB/CLSR composites after cutting them with a razor blade. The CB particles were observed on the surface of the specimen with a CB content ranging from 1 wt% to 3 wt%. Most CB was well distributed in the LSR matrix. However, the aggregation pattern varied depending on the content of CB. In particular, in the specimen with a CB content of 3 wt%, uneven particle distribution was noted compared to the specimens with CB content of 1 wt% and 2 wt%, and some agglomeration was also observed. This accounts for the slight increase in the swelling ratio observed in the CB content of 3 wt%, as mentioned in Section 3.2. Consequently, in the CB/LSR tensile test shown in Figure 4b, the tensile strength increased with the rising CB content but experienced a slight decrease at 3 wt%, which was attributed to the observed agglomeration [38,39].

### 3.4. Compression Set Property

As can be seen in Figure 7 and Table 5, the results of the compression set test exhibit minimal differences at room temperature. However, at 130 °C, an increase in CB content leads to a reduction in the rate of elastic recovery against compression, resulting in a slight increase in the compression set. Specifically, the compression set value at 24 h was 5.43% in neat LSR and 6.81% in CB 3 wt% of LSR. This phenomenon is attributed to the increased crosslinking density of the LSR as CB content rises, actively participating in the crosslinking of the rubber chain. Simultaneously, the movement of the silicone rubber chain is constrained, leading to an increase in stiffness. Consequently, under a continuous load, the number of crosslinks resisting deformation rises, but so does the number of crosslinks broken during the resistance. This results in a lowered crosslinking ratio responsible for deformation recovery, preventing the full recovery of the original thickness. Therefore, the compression set value increases, albeit subtly, with the increment in CB content.

### 3.5. The Mechanism of CB Effect on LSR

In the tensile tests, we confirmed that with the increase in CB content added to LSR, the mechanical property (modulus) increases and the strain rate decreases. Additionally, in a tensile test under deteriorated conditions, as the CB content increased, we observed that LSR composites showed an improvement in resistance to deteriorated conditions. From the results of the swelling test, it was observed that when CB was added to LSR, a low swelling ratio was observed at an appropriate content (2 wt%), demonstrating excellent oil resistance performance. Furthermore, in the compression set test, as the content of CB increased, there was in increase in the cross-linking of the rubber chain. This led to an increase in the number of broken cross-links and a reduction in the cross-linking ratio responsible for deformation recovery, thereby hindering the recovery of the original thickness.

The mechanism behind the excellent mechanical properties and low swelling ratio of CB-added LSR can be elucidated by the physical interaction between the surface of CB and the LSR chain, as well as the chemical bonding in which the OH group of CB reacts with the hydrogen of LSR (Figure 8) [40]. The chemical bonding results in the secure attachment of CB to the LSR chain. The carbon black chemically bonded with LSR can then surround the rubber molecular chain and cause physical crosslinking, roughly as shown in Figure 9a [41]. Essentially, the tightly bound CB, integrated into the crosslinking of LSR, enhances the rigidity of CB/LSR and restricts the mobility of the rubber chain, thereby impeding the penetration of oil between the rubber chains, as illustrated in Figure 9. This, in turn, increases the resistance to oil.

## 4. Conclusions

In this study, we investigated the mechanical properties and swelling characteristics of the CB and LSR composite, considering variables such as CB content, time, and temperature. The mechanical property test results demonstrated that with an increase in CB content, both the tensile modulus and deterioration resistance improved, which was attributed to the strong bonding between CB and the LSR chain effectively delaying the deterioration caused by oil. In the same vein, the restricted mobility of the chain led to a reduction in the swelling ratio for oil. Although there was a slight increase in the compression set with the addition of CB, it was not significant. These findings suggest the potential utility of CB in LSR products exposed to the oil, such as gaskets and seals. However, it is crucial to note that at a relatively high CB content (3 wt%), mechanical properties and oil resistance slightly decreased due to the dispersion and agglomeration of the CB. Therefore, future efforts should focus on improving the surface treatment and dispersion methods of CB during the production of CB/LSR composites to enhance their acceptability.

## Figures and Tables

**Figure 1 polymers-16-00933-f001:**
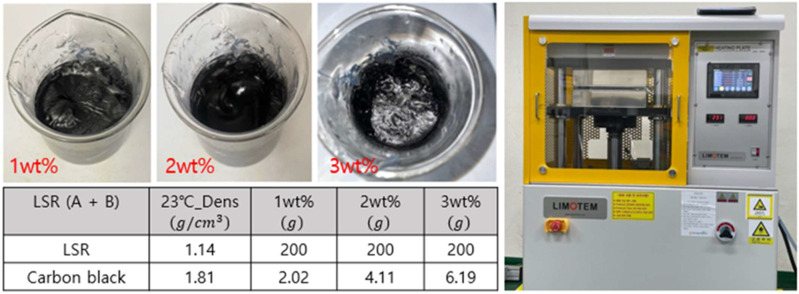
The preparation process of the CB/LSR composite sample.

**Figure 2 polymers-16-00933-f002:**
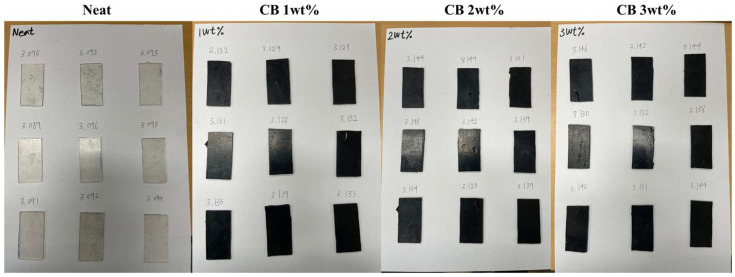
Swelling test specimens.

**Figure 3 polymers-16-00933-f003:**
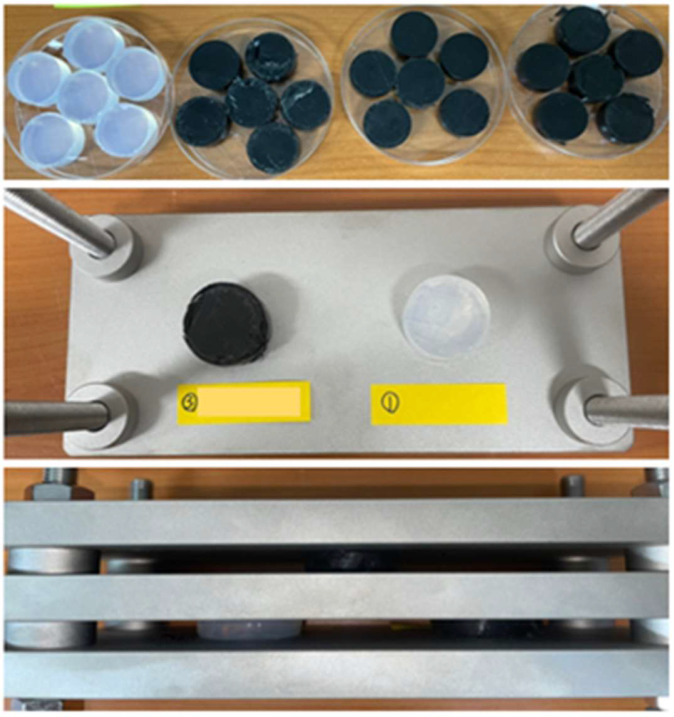
Compression set test jig and specimens.

**Figure 4 polymers-16-00933-f004:**
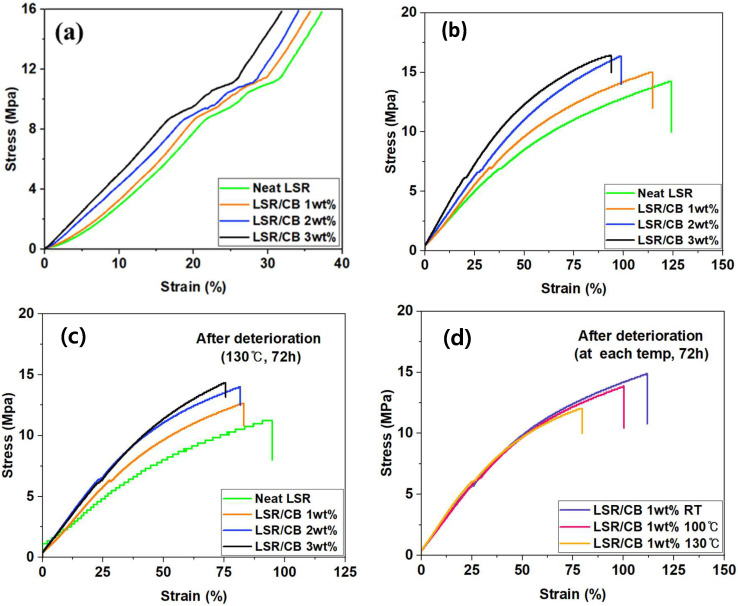
(**a**) Compression curve of CB/LSR, (**b**) tensile curve of CB/LSR, (**c**) tensile curve of CB/SLR after accelerated deterioration (130 °C, 72 h), (**d**) tensile curve of CB/SLR after accelerated deterioration (1 wt%, 72 h) with different temperatures.

**Figure 5 polymers-16-00933-f005:**
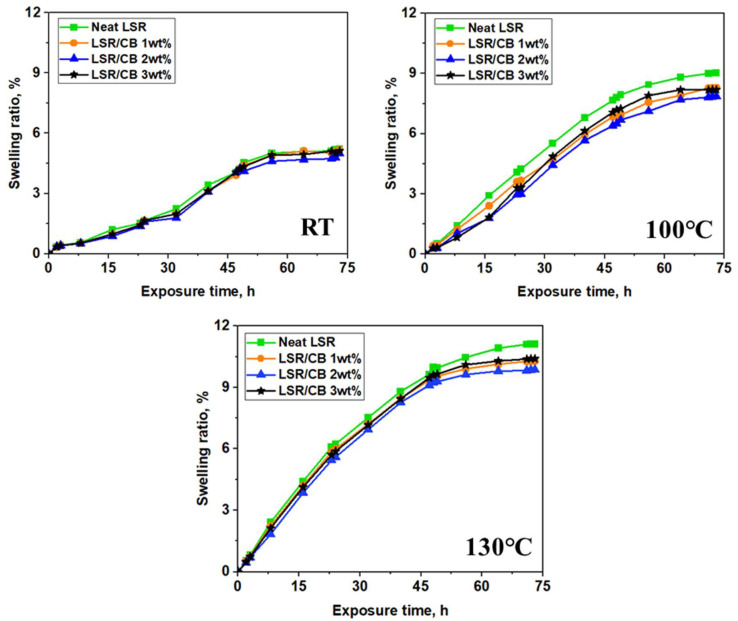
Swelling ratio with time for different temperature of CB/LSR.

**Figure 6 polymers-16-00933-f006:**
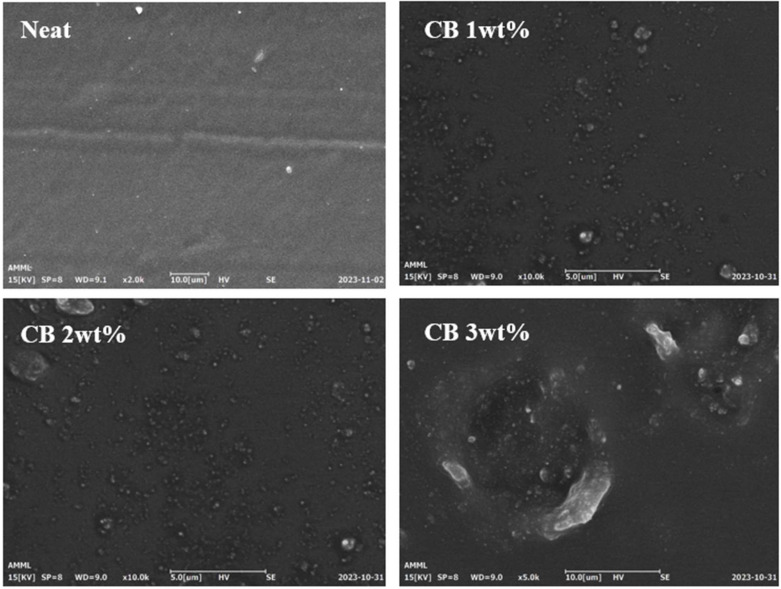
SEM images of CB/LSR specimens.

**Figure 7 polymers-16-00933-f007:**
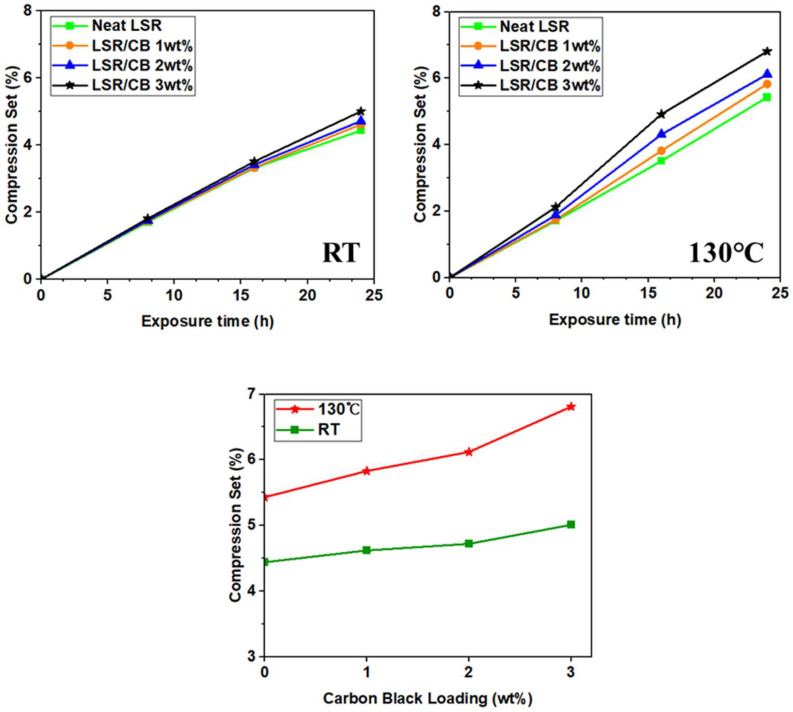
The effect of CB loading on the compression set test of CB/LSR.

**Figure 8 polymers-16-00933-f008:**
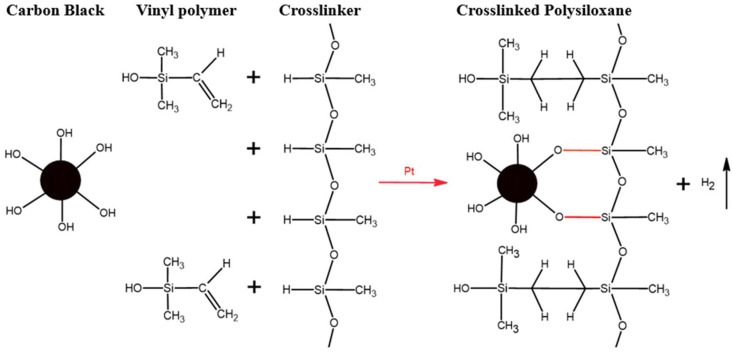
The chemical reaction mechanism between LSR and CB.

**Figure 9 polymers-16-00933-f009:**
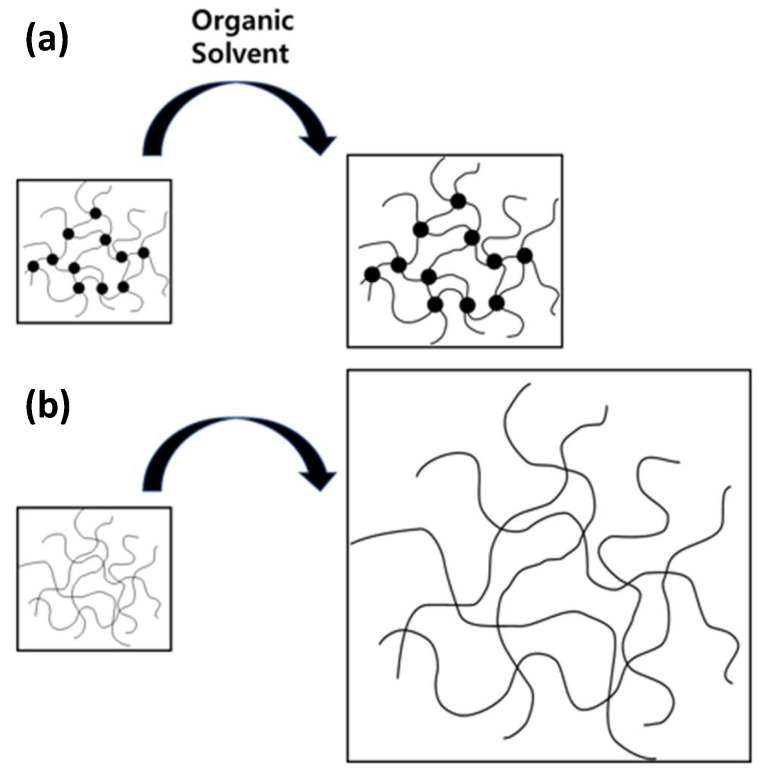
Scheme of the physical network structure LSR swelling with (**a**) and without (**b**) the addition of carbon black.

**Table 1 polymers-16-00933-t001:** Deterioration rate of CB/LSR after accelerated deterioration with ultimate tensile strength (130 °C, 72 h).

Material	Ultimate Tensile Strength-before Deterioration (MPa)	Ultimate Tensile Strength-after Deterioration (MPa)	Deterioration Rate (%)
Neat	14.22	11.21	21.67
CB 1 wt%	14.99	12.62	15.81
CB 2 wt%	16.34	14.57	10.83
CB 3 wt%	16.35	14.32	12.42

**Table 2 polymers-16-00933-t002:** Deterioration rate of CB/LSR after accelerated deterioration with tensile modulus (130 °C, 72 h).

Material	Tensile Modulus—before Deterioration (MPa)	Tensile Modulus— after Deterioration (MPa)	Deterioration Rate (%)
Neat	16.53	13.35	19.24
CB 1 wt%	19.79	17.72	10.46
CB 2 wt%	24.05	22.48	6.53
CB 3 wt%	29.03	23.69	18.39

**Table 3 polymers-16-00933-t003:** Weight ($M_{t}\left(g\right)$) change over oil exposure time of CB/LSR.

Time (h)		0	8	16	24	32	40	48	56	64	72
Neat	RT	3.095	3.112	3.133	3.146	3.164	3.201	3.226	3.250	3.252	3.256
	100	3.092	3.135	3.182	3.223	3.262	3.302	3.334	3.353	3.364	3.371
	130	3.093	3.167	3.229	3.286	3.326	3.365	3.401	3.417	3.431	3.437
CB 1 wt%	RT	3.132	3.149	3.163	3.184	3.194	3.229	3.254	3.285	3.289	3.294
	100	3.129	3.289	3.204	3.244	3.276	3.315	3.343	3.348	3.377	3.389
	130	3.127	3.196	3.258	3.314	3.352	3.392	3.424	3.437	3.444	3.449
CB 2 wt%	RT	3.144	3.160	3.171	3.194	3.200	3.241	3.271	3.289	3.292	3.302
	100	3.149	3.182	3.297	3.240	3.289	3.306	3.354	3.373	3.391	3.397
	130	3.151	3.209	3.272	3.327	3.369	3.411	3.442	3.452	3.459	3.462
CB 3 wt%	RT	3.146	3.163	3.178	3.198	3.206	3.240	3.273	3.302	3.306	3.307
	100	3.142	3.168	3.199	3.246	3.294	3.335	3.370	3.390	3.396	3.397
	130	3.144	3.211	3.274	3.326	3.367	3.409	3.427	3.462	3.467	3.471

**Table 4 polymers-16-00933-t004:** Swelling ratio with different temperature of CB/LSR.

Material			RT 72 h		100 °C 72 h		130 °C 72 h	
**Neat**	$M_{0}$ (g)	$M_{t}\;\left(g\right)$	3.095	3.256	3.092	3.371	3.093	3.437
	**swelling ratio (%)**		**5.202**		**9.238**		**11.121**	
**CB 1 wt%**	$M_{0}$ (g)	$M_{t}\;\left(g\right)$	3.132	3.294	3.129	3.389	3.127	3.449
	**swelling ratio (%)**		**5.172**		**8.309**		**10.297**	
**CB 2 wt%**	$M_{0}$ (g)	$M_{t}\;\left(g\right)$	3.144	3.302	3.149	3.397	3.151	3.462
	**swelling ratio (%)**		**5.025**		**7.876**		**9.858**	
**CB 3 wt%**	$M_{0}$ (g)	$M_{t}\;\left(g\right)$	3.146	3.307	3.142	3.397	3.144	3.471
	**swelling ratio (%)**		**5.118**		**8.116**		**10.407**	

**Table 5 polymers-16-00933-t005:** Compression set with different temperature of CB/LSR.

Material			RT 72 h		130 °C 72 h	
**Neat**	$t_{0}$ (cm)	$t_{1}\;\left(\mathrm{cm}\right)$	12.471	12.450	12.469	12.444
	**Compression set (%)**		**4.44**		**5.43**	
**CB 1 wt%**	$t_{0}$ (cm)	$t_{1}\;\left(\mathrm{cm}\right)$	12.454	12.433	12.452	12.426
	**Compression set (%)**		**4.62**		**5.83**	
**CB 2 wt%**	$t_{0}$ (cm)	$t_{1}\;\left(\mathrm{cm}\right)$	12.478	12.455	12.481	12.452
	**Compression set (%)**		**4.72**		**6.12**	
**CB 3 wt%**	$t_{0}$ (cm)	$t_{1}\;\left(\mathrm{cm}\right)$	12.499	12.474	12.503	12.469
	**Compression set (%)**		**5.01**		**6.81**	

## Data Availability

Data are contained within the article.

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
