# Peer review of "Effects of Carbon Black on Mechanical Properties and Oil Resistance of Liquid Silicone Rubber"

_polymers, 2024, doi:10.3390/polym16070933_

Round 1

Reviewer 1 Report

Comments and Suggestions for Authors

This is an interesting manuscript. However, some corrections are recommended.

1. A term "short SiH" in line 41 should be corrected.

2. The name "Stuber" in line 62 should be corrected.

3. A definition of "DC resistance" (see line 63) should be provided.

4. How and how long both parts A and B of silicone rubber (SR) were mixed ?

5. I my opinion a main reason of agglomeration of CB nanoparticles in SR matrice seems to be insufficient mixing of all components of SR blends. I suggest to improve and to extend mixing of all ingredients and to check it by SEM method.

6. The term "crosslinking density of the CB" (in line248) is incorrect and it should be correcteed. The crosslinking density concerns whole of SR mixes which were vulcanized.

Author Response

  1. A term "short SiH" in line 41 should be corrected.

> Thank you for pointing out the mistake, suggested correction has been made, please check page no.1, line no.40-41. And we added the correspond reference [4].

  1. The name "Stuber" in line 62 should be corrected.

> Thank you for your comments, the suggested correction has been made. Please check page no.2 Line no.61-62.

  1. A definition of "DC resistance" (see line 63) should be provided.

> Thank you for your comments. Suggested correction has been made. Please check page no.2, line no.63-64.

  1. How and how long both parts A and B of silicone rubber (SR) were mixed?

> Part A and B of Liquid silicone rubber (LSR) were mixed for 10 minutes by using mechanical stirrer (same conditions as reference [23]). The information has been added in the manuscript, please check page no.3, line no.108-109.

  1. I my opinion a main reason of agglomeration of CB nanoparticles in SR matrix seems to be insufficient mixing of all components of SR blends. I suggest to improve and to extend mixing of all ingredients and to check it by SEM method.

> Thank you for your comments. We conducted mechanical stirring for 10 minutes with reference to references [23] and [24]. In reference [23], they mentioned the excessive filler addition, rather than stirring time, as the primary factor contributing to agglomeration, also in reference [24], dispersion was ensured by stirring within 10 minutes after adding particles. As mentioned in the conclusion in this manuscript, for our follow-up study, we are looking for a way to improve the dispersion through surface treatment.

  1. The term "crosslinking density of the CB" (in line248) is incorrect and it should be corrected. The crosslinking density concerns whole of SR mixes which were vulcanized.

> Thank you for pointing out the mistake. Suggested correction has been made. Please check page no.9 Line no.256.

Reviewer 2 Report

Comments and Suggestions for Authors

This paper represents the mechanical and oil resistance properties of carbon black reinforced liquid silicone rubber composites. The authors have described the oil resistance properties extensively that were not studied previously in silicone rubber-based composites. The flow of the manuscript is fine. I have only a few queries to authors that may be helpful for further improvement.

1.     Authors have mentioned that physical characteristics have important roles towards the rubber reinforcement. However, they did not mention the physical characteristics of the used carbon black.

2.     How did you optimize the curing parameters?

3.     Figure 4a may be incorrect. Why the slopes are irregular within different compressive strains? While compressive strain should be same for all samples, it shows different for different composites.

4.     Please provide experimental support with the reinforcing mechanism presented in Figure 8. Especially, the presence of hydroxyl group in the carbon black should be evidenced.

5.     I suggest performing cross-link density measuring experiments. How does carbon black affect the cross-link density?

6.     There are some carbon black reinforced silicone rubber composites that have some sensing or electronic applications that can be mentioned in the literature.

Author Response

  1. Authors have mentioned that physical characteristics have important roles towards the rubber reinforcement. However, they did not mention the physical characteristics of the used carbon black.

> Thank you for your comment. We’ve mentioned the physical characteristics in page no.2 Line no. 55-57, and we added the contents of electrical properties of CB in page no.2 Line no. 75-86.

  1. How did you optimize the curing parameters?

> Thank you for your comments. The curing parameters were optimized with reference to the KCC SL7270 data sheet. We added the related sentence in the manuscript, please check reference no. page no.3, line no.112-113.

  1. Figure 4a may be incorrect. Why the slopes are irregular within different compressive strains? While compressive strain should be same for all samples, it shows different for different composites

> Thank you for your comment. As you mentioned, the compressive strain of most materials should exhibit a uniform slope in all samples. However, in some silicone rubber, when the strain reaches a critical value after initially exhibiting linear elasticity, irregular nonlinear strain intervals can occur due to factors such as local network destruction [35-37]. Please check page no.5, line no.177-182 and reference no. [35-37]

  1. Please provide experimental support with the reinforcing mechanism presented in Figure 8. Especially, the presence of hydroxyl group in the carbon black should be evidenced.

> Thank you for your comment. We referred to the previous research on matrix enhanced by CB [34,38]. Please check reference [34,38]. Although the matrix is different, it shares the same elastic properties, and other mechanical properties results show similar trends.

  1. I suggest performing cross-link density measuring experiments. How does carbon black affect the cross-link density?

> Although we were not able to directly measure the crosslink density you mentioned, there are several literatures that show that crosslink density is closely related to the compression set of the elastomer. please check reference [38] mentioned in the manuscript.

  1. There are some carbon black reinforced silicone rubber composites that have some sensing or electronic applications that can be mentioned in the literature.

> Thank you for your suggestion. We have mentioned some literature studies in the introduction section of manuscript. Please check page no.2, and line no.75-86.

Round 2

Reviewer 2 Report

Comments and Suggestions for Authors

Please add the surface area of the used carbon black. Other things are well revised and I am happy to recommend for publication in its current form.